# Tuberculosis Vaccines and T Cell Immune Memory

**DOI:** 10.3390/vaccines12050483

**Published:** 2024-04-30

**Authors:** Fei Li, Wenrui Dang, Yunjie Du, Xiaonan Xu, Pu He, Yuhe Zhou, Bingdong Zhu

**Affiliations:** 1State Key Laboratory for Animal Disease Control and Prevention, Lanzhou Center for Tuberculosis Research, Institute of Pathogen Biology, School of Basic Medical Sciences, Lanzhou University, Lanzhou 730000, China; lf@lzu.edu.cn (F.L.); dangwr21@lzu.edu.cn (W.D.); 120220905730@lzu.edu.cn (Y.D.); 220220926440@lzu.edu.cn (X.X.); hep21@lzu.edu.cn (P.H.); zhyuhe2023@lzu.edu.cn (Y.Z.); 2College of Veterinary Medicine, Lanzhou University, Lanzhou 730000, China

**Keywords:** tuberculosis, vaccine, immune memory, T cell, BCG, subunit vaccine

## Abstract

Tuberculosis (TB) remains a major infectious disease partly due to the lack of an effective vaccine. Therefore, developing new and more effective TB vaccines is crucial for controlling TB. *Mycobacterium tuberculosis* (*M. tuberculosis*) usually parasitizes in macrophages; therefore, cell-mediated immunity plays an important role. The maintenance of memory T cells following *M. tuberculosis* infection or vaccination is a hallmark of immune protection. This review analyzes the development of memory T cells during *M. tuberculosis* infection and vaccine immunization, especially on immune memory induced by BCG and subunit vaccines. Furthermore, the factors affecting the development of memory T cells are discussed in detail. The understanding of the development of memory T cells should contribute to designing more effective TB vaccines and optimizing vaccination strategies.

## 1. The Differentiation of Memory T Cells

Tuberculosis (TB), caused by *Mycobacterium tuberculosis* (*M. tuberculosis*), remains a major global health threat. In 2022, it led to 10.6 million new cases and 1.3 million deaths worldwide [1]. Following *M. tuberculosis* infections, alveolar epithelial cells and alveolar macrophages are the first barrier cells to interact with *M. tuberculosis*, playing critical roles in both bacterial dissemination and anti-TB mucosal immunity [2,3]. Before uptake by phagocytic cells, alveolar epithelial cells secrete antimicrobial peptides, defensins, surfactant proteins, and hydrolases, which contribute to controlling *M. tuberculosis* survival [4,5]. Then, alveolar macrophages phagocytose the bacilli, and infected macrophages exert an inflammatory response, recruiting innate immune cells such as natural killer cells (NK cells), neutrophils, and innate lymphoid cells (ILCs)*,* to eliminate or contain *M. tuberculosis.* If innate immunity does not eliminate invaded *M. tuberculosis*, CD8^+^T cells and Th1-type cell-mediated immune responses are activated, to secrete cytokines such as IFN-γ, which further activate the macrophage system to kill the bacilli inside macrophages and ultimately form a granuloma to control *M. tuberculosis* [6,7]. Although Th1 cells play a prominent role in controlling intracellular pathogen infections, Th2 cells and antibodies are also important in combating *M. tuberculosis infection.* Th2 cells mainly release IL-4 and IL-10 and participate in the activation and proliferation of B cells, which assist in humoral immunity by producing antibodies [8]. In general, over 90–95% of individuals infected with *M. tuberculosis* enter latent TB infection (LTBI), chronically harboring the pathogen without complete clearance, while only about 5–10% infected individuals progress to active TB accompanied by the appearance of clinical symptoms [9]. LTBI is a potential source of active TB. About 5–10% of LTBI reactivate in their lifetime and the risk of LTBI individuals developing active TB will increase significantly in populations infected with human immunodeficiency virus (HIV) (Figure 1).

Following *M. tuberculosis* infection or vaccination, naïve T cells are activated and undergo clonal expansion and proliferation, differentiating into effector T cells (T_eff_) and memory T cells (T_M_) [10,11,12]. T_eff_ plays a role in killing target cells and controlling infections. Upon antigen elimination, T_eff_ enters the contraction phase and undergoes death by apoptosis [13]. Only a small subset persists and differentiates into T_M_ subsets [14]. Upon re-stimulation, T_M_ can rapidly differentiate into T_eff_ and exert a recall response [15]. The metabolic requirements for different types of T cells are variable [16,17,18]. The metabolic reprogramming during T cell differentiation determines the development of memory T cells [19]. T_eff_ rely on aerobic glycolysis to supply energy and synthesize intermediate products, while T_M_ primarily utilize fatty acid oxidation (FAO) and oxidative phosphorylation (OXPHOS) for energy.

T_M_ are phenotypically and functionally heterogeneous [14,20] (Figure 2). T_M_ subsets are mainly divided into effector memory T cells (T_EM_), central memory T cells (T_CM_), stem cell-like memory T cells (T_SCM_), and tissue-resident memory T cells (T_RM_) [10,11,12]. T_CM_ expresses high levels of IL-7 receptor α (CD127), CD62L, and CCR7, while T_EM_ and T_eff_ lack CD62L and CCR7 expression [21,22]. T_CM_ are mainly found in the lymph nodes and survive for over 10 years. Upon antigenic re-stimulation, these cells can rapidly differentiate into T_EM_ and effector cells [21]. T_EM_ mainly exist in the spleen and peripheral nonlymphoid organs and provide short-term protection for about 3 months [23]. IFN-γ is mainly produced by T_eff_ and T_EM_ while IL-2 is predominantly produced by T_CM_ [24]. Moreover, T_SCM_ are defined as the precursors of memory cells and exhibit naïve-like markers (CCR7, CD27, CD127, CD62L). T_SCM_ have high self-renewal and proliferation capabilities and a multi-differentiation potential to generate multiple subsets of memory cells [25,26]. In contrast to other memory T cells, T_RM_ infiltrate local tissues infected by pathogens and stay there long after infection clearance without recycling [27]. CD69 and integrin CD103 are initially critical markers for T_RM_ cells [28,29].

## 2. Variation in Memory T Cells among *M. tuberculosis* Infection

Changes in host immune status are the most direct cause determining the progression of *M. tuberculosis* infection [30]. The immune responses related to LTBI and active TB are dynamic and exhibit divergent patterns. In active TB, antigen-specific T_eff_ play a prominent role in clearing *M. tuberculosis* [31]_._ Due to the high bacterial and antigen load in active TB disease, a large number of antigen-specific T_eff_ become activated. Arrigucci et al. demonstrated that the number of CD4^+^ T cells expressing IFN-γ and TNF-α in active TB was higher than that in LTBI, indicating that active TB was characterized by highly active effector memory Th1 cells [32]. As such, the proportions of IFN-γ and TNF-α in children with active TB are higher than that in children with LTBI [33]. Additionally, in HIV-infected patients with active TB, RD1 antigen-specific CD4^+^ T cells produce high IFN-γ and TNF-α [34]. After successful drug treatment, the bacteria are completely cleared, and some cells survive and differentiate into the memory T cell, which initiates a recall response upon reinfection. For therapeutic TB vaccine, CD8^+^ T cells are activated and differentiate into cytotoxic T lymphocytes (CTLs), which dissolve *M. tuberculosis*-parasitized macrophages by releasing perforin and granulysin.

Tuberculin Skin Test (TST), QuantiFERON-TB Gold (QFT) or interferon-gamma release assays (IGRAs) are common immune sensitization tests for *M. tuberculosis* infections. During *M. tuberculosis* latent infection, the diagnostic results of TST/QFT are generally positive. However, it has been reported that some TST/QFT-positive individuals show a reversal phenomenon (QFT reverters), while some remain persistent positives (Figure 3). QFT and/or TST reversal may reflect the pathogen’s clearance or bacterial load reduction [35,36]. Previous data showed that IGRA positivity reversion rates were 9.4% of household contacts after 3-year follow-up, which increased to 38.2% after 6-year follow-up without previous treatment [37]. In QFT-persistent positive individuals, *M. tuberculosis* may not be cleared, and T cells are activated to perform effector functions, resulting in persistent positive TST and IGRA. Once *M. tuberculosis* is eliminated or reduced to a low bacterial load, the majority of T_eff_ may die by apoptosis, making TST and QFT negative [38,39,40]. 

Latency-associated antigens could induce specific immune memory and play an important role in preventing the reactivation of LTBI. *dosR* and *rpf* are important genes for the survival of *M. tuberculosis* in stress environments. Both dosR regulator and Rv0867c (*RpfA*, which is an important member of the *Rpf* family) could induce the generations of T_EM_ and T_eff_ cells in a long-term latent infection state (ltLTBI) [41,42]. However, compared to ESAT-6 and CFP10, DosR and Rpf antigens exhibit significant monofunctional and bifunctional (IFN-γ and/or TNF-α) T cell responses [43]. In addition, applying peptide microarray technology and sample analysis found that Rv2659c and Rv1738-specific IgA were lower in LTBI individuals than in active TB, while their cellular immune response in LTBI was stronger than that in active TB [44].

Moreover, *M. tuberculosis* persistent stimulation induces the excessive activation of T cells, ultimately leading to T cell exhaustion, which is characterized by progressive loss of effector function and memory T cell potential as well as high and sustained expression of inhibitory immune checkpoint receptors PD-1 and TIM-3 [45,46,47]. Persistent *M. tuberculosis* antigen stimulation induces T cells, including memory precursors and memory T cells, to differentiate into short-lived terminal cells, eventually leading to T cell exhaustion [48]. During 28 months of follow-up in untreated active TB patients, antigen-specific IFN-γ-secreting T cells gradually decreased [49].

## 3. TB Vaccine-Induced Memory T Cells

Bacille Calmette-Guérin (BCG) is the most widely used TB vaccine globally and has been used for more than 100 years. It effectively protects infants and children from miliary TB and meningeal TB but lacks effective protection in adults [50]. It has been suggested that BCG may induce effective protection for 10–15 years [51]. Therefore, developing new TB vaccines that are more effective than BCG or capable of boosting BCG-primed immunity is urgently needed. Novel TB vaccines in clinical trials include live attenuated mycobacterial vaccines, subunit vaccines, viral vector vaccines, inactivated whole cell vaccines, and nuclear acid vaccines, etc. Several TB vaccine candidates are undergoing clinical trials (Table 1).

### 3.1. BCG and Recombinant BCG Vaccine-Induced Immune Memory

The protective immunity of BCG lasts for 10–15 years, possibly because persistent BCG mainly induces T_EM_ [52,53,54]. In mice, BCG-activated CD4^+^ T_EM_ cells secrete IFN-γ and TNF-α, with numbers peaking at 5–6 weeks. In humans, following BCG-priming in infants or children, the frequency of BCG-specific CD4^+^ T cells and the amounts of IFN-γ and TNF-α reach their peak at 10 weeks after vaccination and then gradually decline [55].

Recombinant BCG (rBCG), constructed by overexpressing *M. tuberculosis*-immunodominant antigens in BCG or modifying BCG, is supposed to improve the protective efficacy of BCG. Among them, rBCG ΔureC::hly (VPM1002) is in clinical trials. VPM1002 is an rBCG that overexpresses the listeriolysin gene (*hly*) and has a deleted urease C gene (*ureC*). Since listeriolysin can destroy the phagolysosomal membrane, VPM1002 can effectively activate CD8^+^ and CD4^+^ T cells by interacting with host cell MHC class I and class II molecules to improve BCG antigen presentation. Moreover, it has a shorter survival time in vivo than BCG, promoting the generation of T_CM_ rather than T_EM_ [56,57,58,59,60]. VPM1002 induces the secretion of Th1-type and Th17-type cytokines in mice and provides greater protective effects than BCG [58,59,61]. In an open-label, controlled, randomized, single-administration, dose-escalation phase I clinical trial (NCT00749034), the immunogenicity and safety of VPM1002 were evaluated in QFT^−^, healthy adult male volunteers. It was observed that, 180 days after vaccination, antigen-specific IFN-γ secretion was significantly higher than that when vaccinated with BCG. Meanwhile, VPM1002 could induce the production of IL-2^+^ TNF-α^+^ IFN-γ^+^ CD4^+^ T cells and TNF-α^+^ IFN-γ^+^ CD8^+^ T cells [62]. In another phase II open-label, randomized clinical trial (NCT01479972), healthy newborns without prior exposure to HIV and previous BCG vaccination were vaccinated with VPM1002 and BCG, respectively, to evaluate the safety and immunogenicity in TB-endemic areas. Whole-blood specimens were incubated with BCG or PPD for 7 days at 37 °C to detect IFN-γ production. Compared to pre-vaccination, both VPM1002 and BCG vaccination could induce multifunctional IL-2^+^ TNF-α^+^ IFN-γ^+^ CD4^+^ and TNF-α^+^ IFN-γ^+^ CD8^+^ T cells at 6 months after the last immunization, but there was no significant difference between the two vaccines. Interestingly, IL-17^+^ CD8^+^ T cells were more induced in the VPM1002 group than that in the BCG group [63]. They concluded that the CD4^+^ T cell response induced by VPM1002 vaccination was similar to that induced by BCG, but the CD8^+^ T cell response was superior to BCG.

**Table 1 vaccines-12-00483-t001:** TB vaccine candidates in clinical trials.

Vaccine	Vaccine Composition	Animal Experiment	Clinical Trial	Reference
M72/AS01E	Antigens: Mtb39A, Mtb32A Adjuvant: AS01E, containing immunostimulants MPL and active fraction of Quillaja saponaria (QS21)	In the guinea pig model, protective efficacy lasting over 1 year. As BCG booster vaccine, providing protective efficacy superior to BCG.	Inducing higher numbers of multifunctional CD4^+^ T cells and CD8^+^ T cells, lasting 180 days after the last immunization in healthy population. The protection efficacy of M72/AS01E was 49.7% after 3 years of follow-up in the LTBI population.	[64,65,66,67]
ID93+GLA-SE	Antigens: Rv2608, Rv3619, Rv3620, Rv1813 Adjuvant: GLA-SE, GLA-SE, a synthetic TLR4 agonist GLA formulated in the squalene-in-water stable emulsion	As BCG booster vaccine, significantly reducing the bacterial load, superior to BCG alone in mice. In guinea pigs, as BCG booster vaccine, providing long-term protection against *M. tuberculosis* infection.	Inducing antigen-specific IgG antibody responses and multifunctional CD4^+^ T cell responses, lasting 238 days in healthy adults.	[68,69,70,71,72]
H56:IC31	Antigens: Ag85B, ESAT6, Rv2660c Adjuvant: IC31 being composed of anti-microbial peptide (KLK) and oligodeoxynucleotide (ODN1a)	In cynomolgus macaques’ models, as BCG booster vaccine, reducing pulmonary pathologic changes following *M. tuberculosis* infection.	Inducing a higher frequency of antigen-specific, multifunctional CD4^+^ T cells approximately 100 days after the last immunization in healthy adults.	[73,74,75]
GamTBvac	Antigens: ESAT-6, CFP10, Ag85A Adjuvant: Dextran/CpG	Effectively inducing antigen-specific IFN-γ responses 5 weeks after final immunization in mice. In mouse and guinea pig models, preventing aerosol and intravenous attacks of *M. tuberculosis* H37Rv strain.	Inducing stable and high CD4^+^ T cell responses and IgG responses 83 days after the final immunization.	[76,77,78]
H4:IC31	Antigens: Ag85B, TB10.4 Adjuvant: IC31	As BCG booster vaccine, enhancing BCG-induced memory CD4^+^ T cells and protection in mice.	6 months after the final immunization, BCG-prime and H4:IC31 boosting provides protective efficacy of 30.5%, lower than that in BCG boosting vaccination (45.4%).	[79,80]
AEC/BC02	Antigens: Ag85b, ESAT6 and CFP10 Adjuvant: CpG and aluminum	In guinea pig model of latent infection, reducing the bacterial load in the lungs and spleen.		[81]
MVA85A	Antigen: Ag85A Recombinant poxviral vector: Modified vaccinia Ankara (MVA)	In BCG-vaccinated calves, boost with MVA85A calves showing a wider T cell repertoire than the BCG revaccination groups.	As BCG booster vaccine, inducing long-lasting, polyfunctional *M*. *tuberculosis*-specific CD4+ memory T lymphocyte populations 24 weeks following MVA85A administration. The MVA85A vaccination was well tolerated and immunogenic, but there was no efficacy against *M tuberculosis* infection or disease in adults infected with HIV-1.	[82,83,84,85]
AdHu5Ag85A	Antigen: Ag85A Vector: recombinant replication-defective human serotype 5 adenovirus-vectored (AdHu5-vectored)	As BCG booster vaccine, mucosal boost with AdHu5Ag85A enhancing the antigen-specific T cell responses, improving the survival and bacterial control after challenged with *M. tuberculosis* in rhesus macaques.	Low-dose aerosol immunization eliciting respiratory–mucosal immunity. Both aerosol inhalation and intramuscular injection of AdHu5Ag85A were safe and well tolerated.	[86,87]
TB/FLU-04L	Antigen: Ag85b and ESAT6 Vector: recombinant attenuated influenza strain (Flu NS106)	The TB/FLU-04L intranasal vaccine against TB was safety in mouse, ferrets, monkeys and rabbit model.		[88,89]
VPM1002	rBCG ΔureC::hly	Providing greater protective effects than BCG in mice.	Inducing significantly higher IFN-γ secretion in the previous BCG-uninoculated individuals 180 days after vaccination. Inducing multifunctional CD4^+^ and CD8^+^ T cells 6 months after the last immunization in healthy newborns.	[58,59,61,62,63,90]
RUTI	Detoxified and liposomed, cellular fragments of *M. tuberculosis*	RUTI-treated animals showed lower bacillary load than PBS and BCG groups in the mouse and guinea pig models.	Triggering specific T cell responses against *M. tuberculosis* structural and secreted antigens like PPD, 16 kDa and 38 kDa in healthy volunteers, compared with control subjects.	[91,92,93,94]
MTBVAC	Live, genetically attenuated MTB	In the macaque model, MTBVAC induced similar protective efficacy as BCG 21 weeks after vaccination.	Inducing significantly higher vaccine-specific CD4 and CD8 T cell responses 360 days after vaccination in healthy infants. MTBVAC was at least as immunogenic as BCG 210 days after vaccination in the healthy population.	[58,59,61,62,63,95,96,97]
DAR-901	Inactivated whole cell tuberculosis booster vaccine	Among animals primed with BCG, boosting with DAR-901 at 1 mg provided greater protection against aerosol challenge than a homologous BCG boost in mouse model.	DAR-901 recipients exhibited increased DAR-901 antigen-specific polyfunctional or bifunctional T cell responses compared to baseline. A three-dose series of 1 mg DAR-901 was safe and well-tolerated but did not prevent initial or persistent IGRA conversion.	[98,99,100]

GLA, glucopyranosyl lipid adjuvant; SE, stable emulsion; TLR, Toll-like receptor; TDB, trehalose-6,6-behenate; CpG, cellular guanine phosphate; ODN1a, oligodeoxynucleotide (ODN) 1a.

### 3.2. TB Subunit Vaccine-Induced Immune Memory

The TB subunit vaccine consists of immunologically active components such as proteins, peptides and glycolipids from *M. tuberculosis*. It has the advantages of high efficiency, safety and low cost. The adjuvant of subunit vaccine can help release antigens slowly and maintain a long period at low levels, which contributes to inducing the formation of T_CM_ [101,102,103]. It has been proven that mice vaccinated with subunit vaccines H56 [104] and LT69 [103] induce long-lived memory T cells, which may include T_CM_. Subunit vaccines Ag85B-ESAT-6/CAF01 [105] and H56 [104] could induce the secretion of IL-2, TNF-α, IFN-γ and other cytokines, among which IL-2 is beneficial to the differentiation and proliferation of memory T cells [106,107].

Currently, there are six subunit vaccines in clinical trials, including M72/AS01E, GamTBvac, H56:IC31 (AERAS-456), H4:IC31 (AERAS-404), ID93+GLA-SE and AEC/BC02 (Table 1).

(1) M72/AS01E: M72/AS01E consists of two highly immunogenic *M. tuberculosis* proteins (Mtb39A and Mtb32A) and the adjuvant AS01E. M72/AS01E has shown promising results in several phase I and II clinical trials in adolescents and adults [64,65,108,109,110,111,112,113]. In a phase II double-blind, controlled, randomized clinical trial (NCT00950612), the safety and immunogenicity of M72/AS01E were evaluated in QFT^−^, HIV-uninfected, BCG-vaccinated adolescents in a TB-endemic area. It was observed that, 180 days after the last immunization, M72 peptide-specific IL-2^+^ IFN-γ^+^ TNF-α^+^ CD4^+^ T cells and IFN-γ^+^ TNF-α^+^ CD8^+^ T cells were higher than at pre-vaccination. T cell responses induced by M72/AS01E vaccination were much higher in *M. tuberculosis*-infected individuals than in QFT^−^ individuals. Additionally, the difference between the two groups gradually decreased over time [64]. The reason for this may be that *M. tuberculosis*-infected individuals have already induced *M. tuberculosis* antigen-specific T_CM_ or T_EM_ in their body. After M72/AS01E vaccination, these T cells can be activated and exert a recalling response, which is consistent with the results of another phase IIa clinical trial (NCT00600782) [112]. The protective effect of M72/AS01E was evaluated in a multicenter, double-blind, randomized, placebo-controlled phase IIb clinical trial (NCT01755598) in BCG-vaccinated, QFT^+^ adults. The morbidity of pulmonary TB was lower in the M72/AS01E-vaccinated group than that in the placebo group. A total of 180 days after the last immunization, the protection efficacy of M72/AS01E was 54% [113]. After 3 years of follow-up, M72/AS01E provided 49.7% protection against the recurrence of active TB, indicating that M72/AS01E could induce the formation of long-lived memory T cells and provide partial protection against LTBI reactivation [65].

(2) ID93+GLA-SE: ID93+GLA-SE is composed of four *M. tuberculosis* antigens (Rv1813c, Rv2608, Rv3619c, Rv3620c) and Toll-like receptor 4 (TLR4) agonist GLA-SE as adjuvant. In a mouse model, ID93+GLA-SE immunization induces CD4^+^ T cells to secrete high levels of IFN-γ, IL-2 and TNF-α [68,114]. ID93+GLA-SE boosting BCG significantly reduced the bacterial load following *M. tuberculosis* challenge, superior to BCG alone [68]. Also, in guinea pig models, the ID93+GLA-SE vaccine could boost BCG-induced response and provide long-term protection against *M. tuberculosis* infection [69].

In a randomized, double-masked, dose-escalation phase I clinical trial (NCT01599897), the safety and immunogenicity of ID93+GLA-SE were evaluated in BCG-unvaccinated, HIV-negative, QFT^−^ and healthy adults. ID93 antigen peptide-specific multifunctional CD4^+^ T cell responses with increased IFN-γ, IL-2 and TNF-α secretions lasted until day 238 [70]. In another randomized, double-masked, placebo-controlled phase I clinical trial (NCT01927159), the safety and immunogenicity of ID93+GLA-SE were measured in HIV-negative, BCG-vaccinated, QFT^−^ and healthy adults from South Africa. Whole blood was stimulated with ID93 antigen to detect the levels of IFN-γ, IL-2, IL-17 and TNF-α secreted by CD4^+^ and CD8^+^ T cells. Compared to the placebo group, ID93+GLA-SE vaccination induced higher antigen-specific, IFN-γ^+^ IL-2^+^ TNF-α^+^ CD4^+^ T cell responses 182 days after the last vaccination, indicating that it induced the production of long-lived memory T cells [71]. Furthermore, a randomized, double-blind, placebo-controlled phase IIa clinical trial (NCT02465216) was conducted in Cape Town, South Africa in BCG-vaccinated, HIV-negative adults. ID93+GLA-SE vaccination (2 μg ID93 + 5 μg GLA-SE) induced antigen-specific, multifunctional IFN-γ^+^ IL-2^+^ TNF-α^+^ CD4^+^ T cell and IgG responses that lasted for 6 months, significantly higher than those in the placebo group, indicating that the vaccine was capable of inducing long-lived memory T cells [72].

(3) H56:IC31: H56:IC31 consists of *M. tuberculosis* early secreted antigen Ag85B, ESAT-6 and latency-associated antigen Rv2660c with IC31 adjuvant. IC31 is a two-component adjuvant of anti-microbial peptide (KLK) and oligodeoxynucleotide (ODN1a), a Toll-like receptor 9 (TLR9) agonist [115]. Incorporating Rv2660c into the fusion protein, which is composed of ESAT-6 and Ag85B, significantly enhances the protective effect of the H56 vaccine against *M. tuberculosis* infection in mice [104]. In a cynomolgus macaque model, H56/IC31, as a BCG booster vaccine, alleviated clinical pulmonary pathologic changes at 64 weeks after *M. tuberculosis* infection and prevented the recurrence of latent infection [73].

In an open-label phase I clinical trial in South Africa (NCT01967134), the safety and immunogenicity of H56:IC31 were first evaluated in *M. tuberculosis*-infected and QFT^−^, BCG-vaccinated healthy adults. Low-dose (15 μg H56, 500 nmol IC31) and high-dose (50 μg H56, 500 nmol IC31) vaccines were intramuscularly administered three times. Approximately 100 days after the last immunization, the low-dose H56 vaccination induced a higher frequency of polyfunctional IFN-γ^+^ TNF-α^+^ IL-2^+^ CD4^+^ T cells compared to pre-vaccination. Meanwhile, the H56-induced T cell responses in the *M. tuberculosis*-infected population were stronger than that in healthy individuals, which may be related to the pre-existing *M. tuberculosis* antigen-specific T cells in *M. tuberculosis*-infected population [74]. In another randomized, open-label phase I/II clinical trial (NCT02503839), H56:IC31 vaccine induced multifunctional CD4^+^ T cells producing antigen-specific IFN-γ, IL-2 and TNF-α lasting for 238 days. This suggests that H56:IC31 induced the formation of long-lived memory T cells [75].

(4) GamTBvac: The antigens Ag85A, ESAT6 and CFP10 are non-covalently bonded to the framework constructed by the glucan binding domain (DBD), and combined with the DEAE-dextran nanoparticle adjuvant containing CpG oligodeoxynucleotides (TLR9 agonist) to form a vaccine. Among them, the DBD framework facilitates antigen extraction and presentation [116]. GamTBvac effectively induced antigen-specific IFN-γ responses in lymph nodes and spleen cells in mice models. Both GamTBvac-prime/boost and BCG-prime/GamTBvac-boost regimens were effective in preventing aerosol and intravenous attacks of *M. tuberculosis* H37Rv strain [76]. In a phase I open-label clinical trial (NCT03255278), conducted in Russia (Moscow region), the safety and immunogenicity of different doses of GamTBvac were evaluated in QFT^−^, BCG-vaccinated, healthy adults. After whole blood was stimulated with recombinant antigens DBD-Ag85a and DBD-ESAT6-CFP10 for 72 h, compared with the pre-vaccination period, the half-dose group (DBD-ESAT6-CFP10, 12.5 μg; DBD-Ag85a, 12.5 μg) induced stable and high CD4^+^ T cells responses and IgG responses 83 days after the final immunization. GamTBvac has the capacity to induce the secretions of TNF-α, IP-10, IL-17 and IL-9 [77]. Another Phase II double-masked, randomized, multicenter, placebo-controlled clinical trial (NCT03878004) showed that GamTBvac vaccination induced a high antigen-specific CD4^+^ T cell response compared to pre-vaccination at 93 days after the last immunization [78].

(5) H4:IC31: H4:IC31 is formed by Ag85B-TB10.4 (H4) antigen with adjuvant IC31. Currently, H4:IC31 is mainly used as a BCG booster vaccine. H4:IC31 enhanced BCG-induced memory CD4^+^ T cells and protective efficacy in mice [79]. However, in a randomized, three-arm, placebo-controlled, partially blinded phase II clinical trial (NCT02075203) in HIV-uninfected, BCG-vaccinated, QFT^−^ adolescents, the protective efficacy of H4:IC31 vaccine (30.5%) was lower than that of BCG vaccination (45.4%) [80].

(6) AEC/BC02: AEC/BC02 vaccine, developed by the China Academy of Food and Drug Administration (CAFDA), Beijing, China, comprises Ag85b, ESAT6-CFP10 antigen with the adjuvant BC02 (consisting of CpG with aluminum) [81]. The synergistic effect of CpG and aluminum induces a robust Th1 immune response [117]. In the guinea pig model of latent infection, AEC/BC02 protected the guinea pigs from disease progression, effectively controlled the reactivation of *M. tuberculosis*, and reduced the bacterial load in the lungs and spleen [81]. Currently, two phase I clinical trials (NCT03026972, NCT04239313) about AEC/BC02 have been conducted in PPD-negative and IGRA-negative volunteers with unpublished results. There is also an ongoing phase II clinical trial (NCT05284812) in LTBI individuals.

(7) Other pre-clinical subunit vaccines: In preclinical studies, some vaccines have also shown good protective effects. Among them, LT70 is a multistage subunit vaccine composed of immunodominant antigens of *M. tuberculosis* (Ag85B, peptide 190–198 of MPT64, ESAT-6, latency-associated antigen Rv2626c and proliferative phase antigen Mtb8.4) and the adjuvant DDA+Poly(I: C) [118]. In mice models, compared to the PBS and BCG groups, the LT70 subunit vaccine could induce a Th1-type immune response, producing high levels of IFN-γ, IgG2c and IgG1 antibodies. The protective efficacy of LT70 against *M. tuberculosis* infection in mice was superior to BCG at 30 weeks after the last immunization, suggesting that LT70 has the ability to induce the formation of long-lived memory T cells [118]. 

The CMFO/DMT subunit vaccine is fused with four proteins (Rv2875, Rv3044, Rv2073c and Rv0577). Except for Rv0577, the others are latency-associated antigens. In a mouse latent infection model, CMFO-DMT was effective in preventing *M. tuberculosis* reactivation by eliminating bacterial load in the lungs and spleen, suggesting that CMFO-DMT is a promising TB vaccine candidate that can prevent the reactivation of LTBI [119].

### 3.3. Virus-Vectored Vaccine-Induced Immune Memory

Viral vector vaccines rely on recombinant viruses to deliver antigens without exogenous adjuvants [120]. Recombinant viral vector vaccines can mimic the invasion of pathogens, thereby triggering a robust immune response. These vaccines contain viral vectors that harbor exogenous antigenic segments, allowing them to invade host cells and replicate extensively within them. As a result, they can elicit significant cellular and humoral immune responses without exogenous adjuvants and have the potential to enhance antigen-specific immune memory. After the activation of the host’s immune response, commonly used genotoxic or replication-deficient viruses are swiftly eliminated. Subsequently, antigen-specific immune cells gradually transform into memory cells and can persist for long periods [121].

Currently, viral vectors used in TB vaccine research include influenza virus, Sendai virus (SeV), adenoviruses (Ad), poxviruses, lymphocytic choriomeningitis virus (LCV), cytomegalovirus (CMV), lentiviruses and vesicular stomatitis virus (VSV). Among them, some viral vector vaccines have entered clinical trials, such as AdHu5Ag85A (NCT02337270 I), TB/FLU-01L (NCT03017378 I), TB/FLU-04L (NCT02501421 I), ChAdOx1.85A (NCT03681860 IIa) [122] and MVA85A (NCT00953927 IIb) [123].

AdHu5Ag85A, formerly known as Ad5Ag85A, is a recombinant human type 5 adenovirus (AdHu5) expressing *M. tuberculosis* Ag85A antigen [124]. A phase I clinical trial was conducted on 31 BCG-vaccinated, 18–55 years old healthy adults in Canada (NCT02337270), and bronchoalveolar lavage fluids were collected at 2 weeks and 8 weeks post-vaccination to assess the immunogenicity of AdHu5Ag85A. The results showed that AdHu5Ag85A aerosol immunization induced the production of antigen-specific T cells with respiratory mucosal homing and T_RM_ properties [86].

MVA85A is a poxviruses virus-vectored vaccine, expressing *M. tuberculosis* antigen Ag85A. It uses the Ankara strain of the vaccinia virus as a vector to induce a T cell immune response. Although MVA85A induces long-lasting Ag85A-specific T cell responses in immunocompetent individuals, it failed to enhance BCG-primed protective efficacy in infants [123]. ChAdOx1.85A is a simian adenovirus vector-based TB vaccine, which expresses the MTB antigen Ag85A [125]. ChAdOx1.85A induces polyfunctional CD4^+^ T cells (IFN-γ, TNF-α and IL-2), IFN-γ^+^ TNF-α^+^ CD8^+^ T cells, and Ag85A-specific IgG responses, which can be boosted by MVA85A [122].

SeV85AB is a novel Sendai virus (SeV) vector vaccine that expresses *M. tuberculosis* antigens Ag85A and Ag85B [90,126]. The vaccine SeV986A is a recombinant SeV viral vector that expresses three latency-associated antigens (Rv2029c, Rv2028c and Rv3126c) and Ag85A [127]. In a mouse model, SeV986A immunization via the intranasal route induces higher immunogenicity compared to intramuscular injection.

A novel vesicular stomatitis virus (VSV-846) expressing TFP846 (Rv3615c-Mtb10.4-Rv2660c) [128] induces an effective antigen-specific T cell response through a single intranasal injection, effectively limiting bacterial growth. Mice immunized with VSV-846 show long-term protection against *M. tuberculosis* infection compared to the mice vaccinated with p846 or BCG. An increase in memory T cells is also observed in the spleens of VSV-846-vaccinated mice at 24 weeks.

## 4. Factors Affecting Immune Memory Induced by TB Vaccines

### 4.1. TCR Diversity, Antigen Dose and Route of Inoculation

Antigen dose, stimulation duration and vaccination route affect T_M_ differentiation. T cell antigen receptors (TCRs) signal is an important determinant of T-lymphocyte differentiation [129]. The diversity, affinity and subsequent strength of the TCR signal determine the heterogeneity of T cells [130,131,132]. Low-avidity TCR is critical for protection from heterologous reinfections [133]. Strong TCR signals drive T cell terminal differentiation, while weaker signals induce T cell differentiation toward the memory pattern (Figure 1). For instance, in studies of influenza virus infection, short-term antigenic stimulation favored T_CM_ generation, while sustained antigenic stimulation favored T cell differentiation toward T_EM_ or even T_eff_ [134]. Studies of TB subunit vaccines found that low doses such as 0.5–1 μg H56 [104] and 2 μgLT69 [103] immunizing mice induced longer-lasting immune protection.

Intranasal inoculation with attenuated strains of herpes simplex virus promotes T_EM_ production more than a peritoneal injection, probably due to the activation of mucosal immune response via the intranasal route [135]. Meanwhile, the T_RM_ response in lung tissue can be enhanced through intranasal administration [53]. Moreover, intravenous injection of BCG vaccine in non-human primates has a better protective effect than subcutaneous injection [136].

### 4.2. Immunization Strategies

Subunit vaccines have poor immunogenicity and require multiple booster immunizations to enhance the long-lasting immune memory response [67,79,125,137]. The vaccine immunization interval is the main factor affecting the prime-boost vaccination. It is believed that the optimal time for boosting would be between the late stage of T cell expansion and the maintenance period, aiming at inducing a longer-lasting immune memory [138]. If the interval between prime and booster is short, T_eff_ are predominantly induced and T_M_ production is lessened. For recombinant viral vector vaccines, a combination of different viral vector types and prime-boost immunization strategies are applied to achieve a greater immune response against *M. tuberculosis* or other pathogens infection [139,140,141]. To explore the effect of vaccination intervals on T cell immune memory, LT70 was immunized at 0–3–6, 0–4–12 and 0–4–24 weeks. The 0–4–12 weeks immunization induced more T_CM_ like cells and could significantly reduce pulmonary bacterial load than that at the 0–3–6 weeks vaccination interval [142], suggesting that prolonged subunit vaccine booster intervals contribute to the induction of T_CM_ and effectively prevent *M. tuberculosis* infection.

TB subunit vaccines are supposed to be used as boosters for BCG to improve BCG-primed protective efficacy. However, traditional short-interval booster immunization regimens usually induce T_EM_ and provide short-term protection [142,143]. To optimize the boosting strategy of subunit vaccines, following BCG priming, both Mtb10.4-HspX (MH) immunizations twice at 12–24 weeks and ESAT6-CFP10 (EC) boosting thrice at 12–16–24 weeks increased the number and function of long-term memory T cells and promoted the protective efficacy against H37Ra infection. This indicated that, after BCG priming, “non-BCG” antigen EC vaccinated thrice and BCG antigen MH boosting twice at appropriate intervals would enhance long-lived memory T cell-mediated immunity [144]. H4-IC31H vaccine boosting at 19 and 22 weeks after BCG immunization significantly reduced bacterial loads in the lungs and spleen compared to that in the BCG group [79]. However, the recent phase II clinical trial in BCG-vaccinated, QFT^−^ adolescents, the protective efficacy of the H4:IC31 vaccine (30.5%) was lower than that of BCG vaccination (45.4%) [80].

### 4.3. Cytokines

The differentiation of memory T cells is regulated by various cytokines, such as IL-7, IL-15, IL-21, et al. Among them, IL-7 promotes T cell differentiation and is particularly crucial for the maintenance of T_CM_ homeostasis [145]. Recombinant adenovirus encoding cytokines IL-7 (rAd-IL-7) significantly promotes TB subunit vaccines LT70 and MH to induce more CD4^+^ T_CM_-like cells compared to the sham control [146]. IL-7 binds to the IL-7 receptor (IL-7R) to activate the Janus kinase (JAK)-STAT and phosphatidylinositol 3-kinase (PI3K)-AKT pathways [147]. The activation of JAK1/3-STAT5 initiates the expression of Bcl-2, which has anti-apoptotic effects and maintains the homeostatic proliferation of memory T cells. STAT5 and PI3K are competitive and work together to maintain the homeostasis of T cell proliferation and differentiation [148]. Unlike the effect of IL-7, IL-15 selectively promotes the proliferation of T_EM_ rather than T_CM_ [23]. The IL-21-mediated STAT3 signaling pathway with anti-inflammatory effects contributes to the maturation of CD8^+^ memory precursor T cells in an inflammatory environment [149].

### 4.4. Transcription Factors

The regulation of T cell memory differentiation by internal and external factors is mostly realized through transcription factors. Many transcription factors have been identified as taking part in regulating the development and differentiation of memory T cells, including Tcf7, kif2, Bach2 [123], Bcl-6, Blimp-1 [150], c-Myc, Id2, Id3 [151], NFAT, NF-κB, Notch1, Notch2, T-bet and STAT3 [152]. Among them, Blimp-1 is associated with the differentiation of short-lived effector T cells [153]. Id2/Id3 deletion leads to loss of effector and memory CD8^+^ T cells, and high Id3 expression predicts T cell differentiation to long-term memory T cells [151]. In patients with STAT3 mutations, the memory T cell-associated transcription factor Bcl-6 is reduced, leading to decreased proliferative and differentiation activity of CD4^+^ and CD8^+^ T_CM_, which makes them susceptible to infection by various viruses, bacteria and fungi. Therefore, STAT3 is also an important transcription factor regulating the formation of immune memory [152]. Adeno-associated virus-mediated IL-7 increases the expression of Id3, Bcl-6 and bach2, which are critical for promoting the generation of long-lived memory T cells [154]. Furthermore, Tcf7 is a downstream transcription factor of the Wnt signaling pathway. Activation of the Wnt/β-catenin signaling pathway promotes the differentiation of CD8^+^ T cells into multipotent memory stem cells (CD44^low^CD62L^high^Sca-1^high^CD122^high^Bcl-2^high^) [154,155].

### 4.5. Drugs

Drugs can regulate the direction of T_M_ differentiation mainly by affecting signaling pathways and metabolism during T cell differentiation. The PI3K-AKT-mTOR signaling pathway and adenosine monophosphate-activated protein kinase (AMPK) pathways are the main pathways regulating T cell differentiation. The mTOR inhibitor (rapamycin) promotes the formation of memory precursors during the expansion phase of T cell response and accelerates the differentiation process of memory T cells during the contraction phase [156]. Adding rapamycin to tumor vaccines [157] and BCG immunization processes can promote the differentiation of T_CM_ [158]. Our laboratory also found that rapamycin enhanced the formation of memory T cells induced by the LT70 subunit vaccine [159]. In addition, the AMPK activator metformin favors the production of memory T cells, suggesting that metabolic alterations regulate the differentiation of memory T cells [160]. Interestingly, pyrazinamide (PZA), a critical first-line drug used in TB therapy, promotes the formation of LT70-induced long-lived memory T cells and improves long-term protective efficacy [161]. The reason for this may be that it reduces pro-inflammatory cytokine production through a peroxisome proliferator-activated receptor (PPAR)-dependent pathway [162].

## 5. Conclusions

Vaccines are powerful weapons for preventing and treating many diseases. Currently, various TB vaccine candidates are in the preclinical development stage or have entered clinical trials, breaking the barrier that the BCG vaccine can only be used for pre-infection prevention. In recent years, vaccine-induced immune memory has received much attention. Maintenance of the memory T cell response after vaccination is a hallmark of immune protection and necessary for long-term protection against re-exposure. Therefore, activating long-term immune memory should be considered in the development of TB vaccines and adjuvants. With the participation of novel adjuvants and the continuous optimization of vaccination strategies, effective TB vaccines can be expected to help achieve the ambitious goal of TB elimination.

## Figures and Tables

**Figure 1 vaccines-12-00483-f001:**
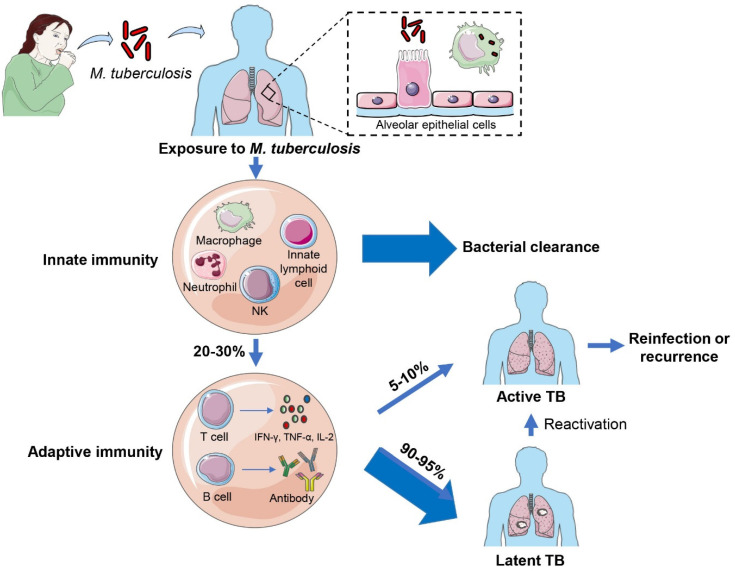
Outcome of *M. tuberculosis* infection. Alveolar epithelial cells are the first barrier cells to interact with *M. tuberculosis*. After *M. tuberculosis* enters the lung by inhalation, alveolar macrophages are activated to engulf the bacteria. Then, infected macrophages exert an inflammatory response, recruiting innate immune cells including NK cells, neutrophils and innate lymphoid cells, to clear the bacteria. If the bacteria are not killed, T cells and B cells are further activated to mediate cellular and humoral immune responses to eliminate *M. tuberculosis*. Among them, approximately 90–95% develop into latent TB infection, and 5–10% will progress to active TB.

**Figure 2 vaccines-12-00483-f002:**
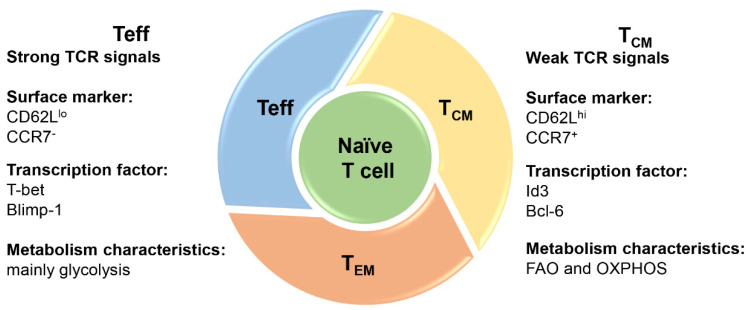
Main subsets and characteristics of effector and memory T cells. Following *M. tuberculosis* infection or vaccination, naïve T cells are activated and undergo clonal expansion and proliferation, differentiating into effector T cells (T_eff_) and memory T cells (T_M_), which mainly include effector memory T cells (T_EM_) and central memory T cells (T_CM_). T cell antigen receptor (TCR) signal is an important determinant of T-lymphocyte differentiation. Strong TCR signals drive T cell terminal differentiation, while weaker signals induce T cell differentiation toward the memory pattern. T_CM_ expresses high levels of CD62L and CCR7, while T_eff_ lacks CD62L and CCR7 expression. For transcription factor regulation, T-bet and Blimp-1 drive the terminal differentiation of T cells, while id3 and Bcl-6 regulate the formation of T_CM_. T_eff_ rely on aerobic glycolysis to supply energy and synthesize intermediate products, while memory T cells primarily utilize fatty acid oxidation (FAO) and oxidative phosphorylation (OXPHOS) for energy.

**Figure 3 vaccines-12-00483-f003:**
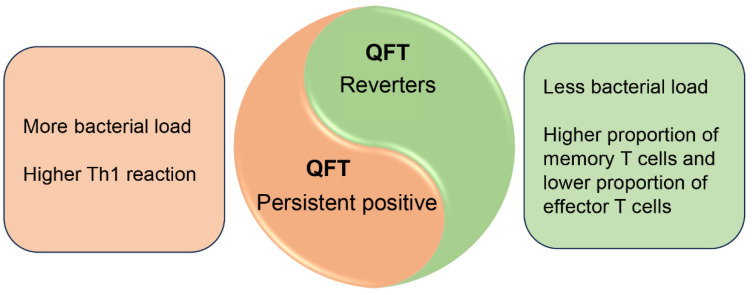
Variation in effector and memory T cells during *M. tuberculosis* latent infection. According to the results of QFT, the LTBI population is mainly divided into two groups: QFT reverters and persistent positives. QFT reverters have a relatively lower bacterial load, a lower proportion of T_eff_ and a greater formation of memory T cells. In contrast, more *M. tuberculosis* bacilli are present in persistent positive populations and continuously stimulate Th1 effector T cells.

## Data Availability

Data sharing not applicable.

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
