# Peer review of "Tuberculosis Vaccines and T Cell Immune Memory"

_vaccines, 2024, doi:10.3390/vaccines12050483_

Round 1

Reviewer 1 Report

Comments and Suggestions for Authors

The topic is timely and important.  The authors review the characteristics of several TB vaccines currently in human trials, and summarize the status of other TB vaccines that are in the pre-clinical phase of development and testing. They explain what is known about the functions of  various subpopulations of memory T cells, and the variables that influence their induction by infection and vaccination. The review is well-written and comprehensive. The cited literature is complete and current.

Comments on the Quality of English Language

The text is generally well-written and the English is quite good. I found few examples of typographical or grammatical errors. 

Author Response

Dear Editor,

Thanks for your careful consideration of our manuscript. We appreciate the helpful comments from reviewers very much. We have made careful revisions accordingly. Our point-by-point responses are as follows.

Reviewer 1

The topic is timely and important. The authors review the characteristics of several TB vaccines currently in human trials, and summarize the status of other TB vaccines that are in the pre-clinical phase of development and testing. They explain what is known about the functions of  various subpopulations of memory T cells, and the variables that influence their induction by infection and vaccination. The review is well-written and comprehensive. The cited literature is complete and current.

The text is generally well-written and the English is quite good. I found few examples of typographical or grammatical errors.

Answer: Thanks for your great comments.

We have revised our manuscript very carefully. If there are any questions, please let us know and we will do more revisions. We are looking forward to hearing from you.

Best wishes,

Fei Li, M. D.

Bingdong Zhu, M. D.

Reviewer 2 Report

Comments and Suggestions for Authors

A bland and not too detailed review, which lists published results without much attempt to tie them together to give a broader understanding of the field.

Author Response

Dear Editor,

Thanks for your careful consideration of our manuscript. We appreciate the helpful comments from reviewers very much. We have made careful revisions accordingly. Our point-by-point responses are as follows.

Reviewer 2

A bland and not too detailed review, which lists published results without much attempt to tie them together to give a broader understanding of the field.

Answer: Thanks for your suggestion. This manuscript mainly reviews the characteristics of several TB vaccines currently in pre-clinical and ongoing clinical trials, especially on the memory responses induced by M. tuberculosis infection and vaccination. Memory T cells are associated with the protective effect induced by vaccines. This review may provide some valuable information in this field. Certainly, we will further explore it in the future. 

We have revised our manuscript very carefully. If there are any questions, please let us know and we will do more revisions. We are looking forward to hearing from you.

Best wishes,

Fei Li, M. D.

Bingdong Zhu, M. D.

Reviewer 3 Report

Comments and Suggestions for Authors

This is a review manuscript to discuss the Tuberculosis vaccine and T cell immune memory

Overall, the manuscript is well presented and focuses on the role of the memory T cell-mediated immune response in enhancing the efficacy of vaccines against M. tuberculosis. There are however some points that should be mentioned to increase the power of the manuscript.

Major concerns

1.1  – The innate immune response induced by type II alveolar epithelial cells is also crucial in the pulmonary control of M. tuberculosis. Authors can explore the role of these cells in the Introduction and Figure 1. - doi.org/10.3390/ijms22052566.

1.2  - Metabolism in CD8+ T cell was either explored r by DOI: 10.1016/j.immuni.2014.06.005

1.3  – Previous data showing IGRA positivity reversion without previous treatment after 6-year follow-up was reported.

1.4  – reference # 57 was not cited correctly. Review. Is missing the reference of VPM1002 phase II in healthy newborns.

1.5  Reference for Table 1 should be included since there are constant changes in the TB vaccine pipeline. https://newtbvaccines.org/tb-vaccine-pipeline/clinical-phase/. Authors may review the data on page 6 lines 195-298. In an update on March 18, 2024, some of the vaccines described have been removed from the pipeline. Authors can keep the vaccines listed and must include data on other vaccines.

1.6  Although MVA85A vaccine failed to enhance the immune response after previous BCG vaccination (ref 97; 2014) new data indicate induction of polyfunctional CD4+ T cells (IFN-γ, TNF-α and IL-2) and IFN-γ+, TNF-α+ CD8+ T cells by ChAdOx1 85A and boosted by MVA85A (2020 and 2024). If the authors consider it important, please include.

1-    Minor concerns

2.1. Remove the word etc on page 1, line 28; page 2, line 47, page 6 line 191, and page 11 line 429. It is not scientifically correct to use it.

2.2. Always use the abbreviation for effector T cells or memory T cell. Ex. page 2, lines 57 and 61 – TM and page 2 line 60 - Teff. Please, review the manuscript.

2.3. Remove the words  - Figure 2 -  from page 2, line 62 and transfer to page 3, as the information provided in this figure is not fully filled in on page 2.

2.5. The section 4.1 should be better explored. It is not clear if vaccine antigen dose is able to induce different antigen stimulation and T cell activation. On the other hand, TCR diversity is clearly associated with vaccine efficacy.

2.6. Regarding references 110-112, describe if the results were from tuberculosis vaccine or other pathogens.

2.7. The results from LT70 were from mice model. Please indicate. There is new data from LT70 inducing TM cells.   

Author Response

Dear Editor,

Thanks for your careful consideration of our manuscript. We appreciate the helpful comments from reviewers very much. We have made careful revisions accordingly. Our point-by-point responses are as follows.

Reviewer 3

This is a review manuscript to discuss the Tuberculosis vaccine and T cell immune memory

Overall, the manuscript is well presented and focuses on the role of the memory T cell-mediated immune response in enhancing the efficacy of vaccines against M. tuberculosis. There are however some points that should be mentioned to increase the power of the manuscript.

Major concerns

1.1  – The innate immune response induced by type II alveolar epithelial cells is also crucial in the pulmonary control of M. tuberculosis. Authors can explore the role of these cells in the Introduction and Figure 1. - doi.org/10.3390/ijms22052566IF: 5.6 Q1 .

Answer: Thanks for your suggestion. Yes, you’re right. Alveolar epithelial cells and alveolar macrophages are the first barrier cells to interact with M. tuberculosis, playing critical roles in both bacterial dissemination and anti-TB mucosal immunity. We have added the role of these cells in the Introduction and Figure 1 to make it more comprehensive.

1.2  - Metabolism in CD8+ T cell was either explored r by DOI: 10.1016/j.immuni.2014.06.005

Answer: Thanks for your suggestion. The metabolic reprogramming during T cell activation and differentiation affects T cell fate and immune responses. We carefully read the above literature and cite these studies in the manuscript to make this review more clearly.

1.3  – Previous data showing IGRA positivity reversion without previous treatment after 6-year follow-up was reported.

Answer: Thanks for your comments. We have carefully read this study and cite the reference (see below) in the manuscript.

[1]Zhang HC, Ruan QL, Wu J, Zhang S, Yu SL, Wang S, Gao Y, Wang FF, Shao LY, Zhang WH. Serial T-SPOT.TB in household contacts of tuberculosis patients: a 6-year observational study in China. Int J Tuberc Lung Dis. 2019 Sep 1;23(9):989-995.

1.4  – reference # 57 was not cited correctly. Review. Is missing the reference of VPM1002 phase II in healthy newborns.

Answer: Thanks for your suggestion. Yes, that reference is not accurate, we have corrected it and added the reference of VPM1002 phase II in healthy newborns in the manuscript.

1.5  Reference for Table 1 should be included since there are constant changes in the TB vaccine pipeline. https://newtbvaccines.org/tb-vaccine-pipeline/clinical-phase/. Authors may review the data on page 6 lines 195-298. In an update on March 18, 2024, some of the vaccines described have been removed from the pipeline. Authors can keep the vaccines listed and must include data on other vaccines.

Answer: Thanks for your great suggestions. As your suggestion, we have added some of the vaccines in Table 1 in the manuscript.

1.6  Although MVA85A vaccine failed to enhance the immune response after previous BCG vaccination (ref 97; 2014) new data indicate induction of polyfunctional CD4+ T cells (IFN-γ, TNF-α and IL-2) and IFN-γ+, TNF-α+ CD8+ T cells by ChAdOx1 85A and boosted by MVA85A (2020 and 2024). If the authors consider it important, please include.

Answer: Thanks for your constructive comments. Yes, new data indicate that ChAdOx1.85A induces polyfunctional CD4+ T cells (IFN-γ, TNF-α, and IL-2), IFN-γ+ TNF-α+ CD8+ T cells, and Ag85A-specific-IgG responses, which can be boosted by MVA85A. We have added these studies to the manuscript.

1-    Minor concerns

2.1. Remove the word etc on page 1, line 28; page 2, line 47, page 6 line 191, and page 11 line 429. It is not scientifically correct to use it.

Answer: Thanks for your suggestions. I have removed the word etc one by one, to make it more accurate.

2.2. Always use the abbreviation for effector T cells or memory T cell. Ex. page 2, lines 57 and 61 – TM and page 2 line 60 - Teff. Please, review the manuscript.

Answer: Thanks for your suggestions. I have corrected the abbreviations one by one.

2.3. Remove the words  - Figure 2 -  from page 2, line 62 and transfer to page 3, as the information provided in this figure is not fully filled in on page 2.

Answer: Thanks for your comments. I have removed the words Figure 2 to Page 3 to make it better.

2.5. The section 4.1 should be better explored. It is not clear if vaccine antigen dose is able to induce different antigen stimulation and T cell activation. On the other hand, TCR diversity is clearly associated with vaccine efficacy.

Answer: Thanks for your suggestions. We have added some detailed information about vaccine antigen doses on immune memory.

TCR diversity affects TCR signal strength, which is involved in CD8+ T cell activation and differentiation. Low-avidity TCR is critical for protection from heterologous reinfections. We have made some revisions to it.

2.6. Regarding references 110-112, describe if the results were from tuberculosis vaccine or other pathogens.

Answer: Thanks for your suggestions. In references 110-112, a variety of pathogens are included, and the original description might be not accurate. We have corrected it to “For recombinant viral vector vaccines, a combination of different viral vector types and prime-boost immunization strategy applied to achieve a greater immune response against M. tuberculosis or other pathogens infection”.

2.7. The results from LT70 were from mice model. Please indicate. There is new data from LT70 inducing TM cells.  

Answer: Thanks for your suggestions. We have indicated the results of LT70 come from a mouse model and have described new data in different sections like 4.2. Immunization strategies, 4.3. Cytokines and 4.5. Drugs in the manuscript.

We have revised our manuscript very carefully. If there are any questions, please let us know and we will do more revisions. We are looking forward to hearing from you.

Best wishes,

Fei Li, M. D.

Bingdong Zhu, M. D.

Reviewer 4 Report

Comments and Suggestions for Authors

This review describes the T cell responses and memory responses induced by Mtb infection and vaccination. It provides valuable information for developing a new vaccine and analyzing its immunity. 

Comments:

1. Can you include T cell response to therapeutic TB vaccine?

2. Please briefly describe the significance of T cell response as a biomarker that correlates with protection.

Author Response

Dear Editor,

Thanks for your careful consideration of our manuscript. We appreciate the helpful comments from reviewers very much. We have made careful revisions accordingly. Our point-by-point responses are as follows.

Reviewer 4

This review describes the T cell responses and memory responses induced by Mtb infection and vaccination. It provides valuable information for developing a new vaccine and analyzing its immunity.

Comments:

  1. Can you include T cell response to therapeutic TB vaccine?

Answer: Thanks for your comments. Yes, we have added T cell response to therapeutic TB vaccine. The T-cell response induced by several therapeutic vaccines, such as ID93+GLA-SE, H56:IC31, and MVA85A, are involved in the manuscript. For therapeutic TB vaccine, CD8+ T cells are also activated and differentiate into cytotoxic T lymphocytes (CTLs), which dissolve M. tuberculosis-parasitized macrophages by releasing perforin and granulysin.

  1. Please briefly describe the significance of T cell response as a biomarker that correlates with protection.

Answer: Thanks for your suggestion. M. tuberculosis is an intracellular pathogen and thus T-cell-mediated responses play a prominent role against TB. Upon vaccination, naÑ—ve T cells are activated and differentiated into effector T cells and memory T cells. Effector T cells play a role in eliminating antigens and controlling infections by secreting cytokines such as IFN-γ and IL-2. Following antigen clearance, the majority of effector cells would die by apoptosis. Only about 5%–10% of T cells persist and continue to differentiate into memory T cell subsets. These memory T cells have long-term survival and retain the capability of homeostatic proliferation. When re-stimulated, memory T cells can generate effector T cells and sustain a recall response, providing a protective effect. Therefore, T cell response can be used as a biomarker associated with protection.

We have revised our manuscript very carefully. If there are any questions, please let us know and we will do more revisions. We are looking forward to hearing from you.

Best wishes,

Fei Li, M. D.

Bingdong Zhu, M. D.
